# Impact of end-stage knee osteoarthritis on perceived physical function and quality of life: A descriptive study from Jordan

**Sumayeh Abujaber** [1]*, **Ibrahim Altubasi** [1], **Mohammad Hamdan**[2], **Raed Al-Zaben**[3]

**1** Department of Physiotherapy, School of Rehabilitation Sciences, The University of Jordan, Amman, Jordan, **2** Department of Special Surgery, Division of Orthopaedics, School of Medicine, The University of Jordan, Amman, Jordan, **3** Department of Orthopaedic Surgery, Royal Medical Services, Amman, Jordan

* s.abujaber@ju.edu.jo

**Data Availability Statement:** All relevant data are within the paper and its Supporting Information files.

## Abstract

### Objective

Of the present study was to evaluate the impact of end-stage knee OA on patient's perception of their functional abilities and quality of life (QoL) using the self-reported questionnaire; the Knee Injury and Osteoarthritis Outcome Score (KOOS), and to determine the contribution of knee pain on patient's perceived outcomes.

### Methods

Patients with end-stage knee OA who are on the waiting list for total knee arthroplasty were recruited in this cross-sectional study. Patients were asked to fill out the KOOS questionnaire. Knee pain for both sides was quantified on a continuous scale from 0–10. Age, and anthropometric data were recorded. Descriptive statistics were calculated for patients' characteristics, and for the scores of each KOOS subscale. Hierarchical linear regression models were created to determine the contributions of knee pain on two KOOS subscales; the function in daily living (KOOS-ADL), and the knee-related quality of life (KOOS-QoL).

### Results

Patients in this study scored low across KOOS subscales (27.7% - 54.2%) with the QoL subscale being the lowest. After accounting for age and BMI, hierarchical linear regressions revealed that knee pain in both sides were determinants of self-perceived KOOS-ADLs, while only knee pain in the most-affected side significantly contributed to lower KOOS-QOL scores.

### Conclusion

End-stage knee OA negatively impact the patients' perceived function and quality of life. Patients' KOOS scores were similar to those reported in other countries, with QoL being the domain most affected. Our findings demonstrate that the level of knee pain has a determinant effect on our patients' perceptions of functional abilities and QoL. As waiting-list

**Funding:** This study was financially supported by the Deanship of Scientific Research at the University of Jordan. This fund was received by SA. https://research.ju.edu.jo/Home.aspx The funders had no role in study design, data collection and analysis, decision to publish, or preparation of the manuscript.

**Competing interests:** The authors have declared that no competing interests exist.

patients, addressing knee pain with a targeted regimen prior to TKA, as well as increasing patient's awareness about knee pain management, may improve/ or minimize deterioration in perceived functional ability and QoL while awaiting TKA.

## Introduction

Osteoarthritis (OA) is a multifactorial, progressive disease that involves degeneration of the hyaline cartilage and surrounding tissue within the joint. OA is the most common joint disorder in adults [1], with the knee being the most affected joint [2]. Knee OA is a highly prevalent musculoskeletal condition, affecting approximately 22.9% of individuals aged 40 and over [3]. Along with hip OA, knee OA is considered the 11th highest contributor to global disability in the elderly [4]. Functional limitations, persistent pain, and poor quality of life are consequences of Knee OA [5, 6].

When knee OA progresses to severe end-stage, and conservative treatment fails to reduce pain and improve the patient's functions, then total knee arthroplasty (TKA) becomes the treatment of choice. In the USA, this surgery was performed for 4.55% of the population ≥50 years of age [7], and the estimated annual incidence of primary TKA will be 3.5 million by 2030 [8]. However, patients with end-stage knee OA may wait for several months before they undergo the TKA procedure due to various reasons. Waiting time for TKA may further deteriorate the patient's condition in terms of pain, physical function, and health-related quality of life [9].

A self-reported questionnaire is a tool that measures the patient's perception regarding their functional abilities and limitations. It allows the patients to express their satisfaction level, physical limitation or symptoms, and quality of life. Self-reported questionnaires are recommended to use as they are easy to administer, inexpensive, and have high internal consistency [10]. Although patients may over- or underestimate their abilities compared to their actual performed physical function, patient perception is certainly an essential part of the management process when describing the patient's functional status and monitoring the outcomes progression. Many disease-specific self-reported questionnaires are available and commonly utilized in OA and TKA populations including Knee Injury and Osteoarthritis Outcome Score (KOOS) [11], Knee Outcome Survey (KOS) [12], and Western Ontario and McMaster Universities Osteoarthritis Index (WOMAC) [13].

Although several studies have examined the functional abilities of patients with knee OA in different countries, the impact of knee OA may vary among different populations due to different lifestyles, everyday activities, and sociocultural differences. Knee OA is a common joint disorder in Jordan, a developing country in the Middle East region. Although studies on its prevalence are lacking, it has been reported that Jordanians have a higher percentage of end-stage knee OA compared to the developed world [14]. Particularly, studies describing the perceived function in Jordanian patients with knee OA are scarce. Jordanians with knee OA showed limited knowledge of the disease pathology and management options that is partly attributed to inappropriate service delivery, lack of education, and cultural factors [15]. These factors could influence the experience of people with knee OA, potentially worsening their pain and joint deterioration. This could ultimately have a negative impact on their functional ability.

Therefore, the purpose of this study is to evaluate the impact of knee OA on patients' perception of functional abilities and quality of life in Jordanians with end-stage knee OA, who are on the waiting list for TKA. The second purpose was to determine how joint pain

influences the patient's perceived functional ability and quality of life. The results from this study will help to characterize how patients perceived their quality of life and functional capacity considering their knee condition. It will also provide evidence regarding the contribution of joint pain to a patient's perception, which might serve to design a targeted rehabilitation program that minimizes functional limitations during the waiting time for TKA.

## Methods

### Study design and participants

This cross-sectional study is part of a longitudinal study that evaluates changes in physical function before and after TKA. Waiting lists for patients with end-stage knee OA scheduled for TKA were obtained from local hospitals. Patients with end-stage knee OA between 50 and 85 years, who are scheduled for unilateral TKA were included. Patients were screened for eligibility using a telephone interview conducted by a research assistant. Exclusion criteria were if patients have other musculoskeletal, neurological, or cardiovascular pathologies that affect their ability to perform daily activities, and if they have a history of cancer in the lower extremity. Moreover, patients who plan to have surgery on their "less affected" knee (staged bilateral TKA) or on other lower extremity joints within a year or had previous arthroplasty surgery less than 1 year were excluded from the study to avoid the potential confounding influence of other joint impairments. Eligible patients were asked to fill out a self-reported questionnaire; the Arabic version of KOOS to evaluate their perceived pain and function. Evaluation session was performed 1 day to several weeks before TKA and took place either at the Physiotherapy department/ School of Rehabilitation Sciences at the University of Jordan or the University of Jordan Hospital. For those who were unable to reach the sitting, a web-based questionnaire was sent to them to fill out online. The study was approved by the Institutional Review Board at the University of Jordan hospital, and patients signed an informed consent form before participation, while those who completed the questionnaire online gave their informed consent by agreeing to answer the questionnaire's questions.

### Participants' characteristics

Age, weight, height, and sex were recorded, and body mass index (BMI) was calculated for each subject. Knee pain, for the affected and less affected sides, was assessed on a continuous scale from 0 to 10, patients were specifically asked to verbally "rate your average pain over the past week from 0 to 10, where 0 is no pain and 10 is the worst imaginable pain".

### Evaluation of symptoms, physical function, and quality of life

The Arabic version of the KOOS was used to evaluate pain, symptoms, physical function, and quality of life [16]. The KOOS is a self-reported tool used to evaluate symptoms and function in patients with knee-related pathologies [11, 17]. The KOOS is a joint- and limb-specific questionnaire, patients in this study provide answers regarding their "most affected" knee only. KOOS includes 42 items across 5 subscales: 1) pain frequency and severity during functional activities (**KOOS-Pain; 9 items**), 2): other symptoms such as the severity of knee stiffness and the presence of swelling (**KOOS-Symptoms; 7 items**), 3) difficulties when performing activities of daily living (**KOOS-ADL, 17 items**), 4) difficulty experienced during sports and recreational activities (**KOOS-Sport; 5 items**), and 5) knee-related quality of life (**KOOS-QOL; 4 items**). Each subscale is scored separately; the answer for each question is rated using a 5-point Likert scale thus scoring from 0 (no problem) to 4 (extreme problem).

A total score for each subscale is calculated and normalized to 100; with 100 indicating no symptoms or difficulties, and 0 indicating severe symptoms or difficulties. This normalization method is used in accordance with the KOOS scoring guideline 2012 (available at www.koos.nu), which calculates the total score as shown in Eq 1.

$$100 - (average\ of\ items/4)*100 \tag{1}$$

Patients were asked to rate the above subscales based on the previous week. The KOOS has been shown to have excellent reliability and good content and construct validity when used for follow-up of a knee injury, including OA [10]. Alfadhel et al. (2018) translated and cross-culturally adapted the original KOOS questionnaire into Arabic in Saudi Arabic patients with knee OA. The psychometric properties of the KOOS's Arabic version were evaluated in that study, and the results showed that the Arabic version of the KOOS is well accepted and has excellent reliability, internal consistency, and construct validity [16]. The KOOS has similar items as the WOMAC physical function subscale with the addition of questions about sport and recreation and knee-related Quality of life.

## Data analysis

Descriptive statistics (mean and SD) were calculated for patients' characteristics (age, height, weight, BMI, and knee pain), and each KOOS subscale. The median for each item's score within each subscale was calculated to indicate the perceived difficulty/severity for that item (based on Likert score), and the percentage of patients who rated that item with a given Likert point was calculated too.

To determine the influence of knee pain (independent variable), measured by KOOS pain-subscale, on the functional abilities measured by KOOS ADLs-subscale and on the quality of life measured by KOOS-QOL (dependent variables), hierarchical linear regression models were created. Separate regression models were created for each dependent variable. Subject characteristics (age and BMI) were potential confounders to the relationships between pain and functional outcomes. In these regression models, age and BMI were first entered into the models. Because pain in the less affected side may affect the dependent outcome, it was added in the second step in the hierarchical regression (this was assessed using the 0 to 10 continuous scale), followed by the addition of knee pain of the affected side reported in KOOS-Pain subscale in the third step. The change in $R^2$ informs us whether the addition of the variable in each step provides significant additional predictive information when accounting for the variance explained by the variables in the preceding steps. The significance of each model, as well as the significant change in $R^2$ between each step, was recorded. Statistical analyses were conducted using the Statistical Package for the Social Sciences (IBM SPSS 23.0, Chicago, IL). Significance level was set at 0.05.

## Results

This study sample included 52 participants. We were only able to recruit four male participants out of the 52 participants, who were drawn from a population with a high female-to-male ratio. The data from the four male participants was included in this analysis, as we did not originally plan to exclusively focus on female patients. Additionally, we conducted the analysis with and without the male participants data and found that results remained unchanged.

The mean age of the sample was 67 years old (range 50–78). The anthropometric characteristics of the patients were summarized in Table 1. Patients expressed a higher level of pain on the "most affected" side (8/10) compared to that on the "less affected" side (4/10) as measured using the 0–10 continuous scale (Table 1).

**Table 1. Anthropometric characteristics, pain and KOOS scores (N = 52).**

|  | Mean | SD | Range |
|---|---|---|---|
| Height (cm) | 156.05 | 7.81 | 125–172 |
| Weight (Kg) | 82.15 | 13.85 | 41–129 |
| BMI (Kg/m$^2$) | 33.67 | 5.00 | 23.4–53 |
| Age (yrs.) | 66.62 | 6.68 | 50–78 |
| Knee Pain "most affected" (0–10) | 7.96 | 1.73 | 3–10 |
| Knee Pain "less affected" (0–10) | 4.15 | 2.91 | 0–10 |
| KOOS-symptom (%) | 54.20 | 20.70 | 10.71–92.86 |
| KOOS-pain (%) | 45.75 | 20.16 | 2.78–84.38 |
| KOOS-ADL (%) | 47.05 | 20.14 | 4.69–91.67 |
| KOOS-QOL (%) | 27.76 | 16.71 | 0–87.5 |

Patients in our sample reported impairments and limitations on the KOOS. The KOOS subscales' scores ranged between 27.7% - 54.2%, with the quality-of-life subscale having the lowest score. The KOOS-Sport subscale was not quantified as most patients did not answer its questions or indicated that it does not apply to them. Table 1 shows the mean and standard deviation of the KOOS subscales.

The median for each item within a specific subscale is presented in Table 2. Regarding the KOOS-Symptom subscale, the items' median ranged from 0–4. Item S5 "Can you bend your knee fully?" showed the highest median (4) with 61.6% of patients responding with "rarely" or "never able to do so". The median on the other items ranged between 0–2.

In the KOOS-Pain subscale, items' median ranged between 1–4, P1 item *"How often do you experience knee pain?"* showed the highest median (4), with 94.2% of patients responding with "daily" or "always". This was followed by the P6 item *"Going up or down stairs"* which showed a median of 3, with 59.6% of patients experiencing "severe" to "extreme pain" when performing that task.

Regarding the KOOS-ADL, the elements with the highest median were for: A8 *"Going shopping"* and A16 *"Heavy domestic duties (moving heavy boxes, scrubbing floors, etc)"* in which 50% and 42.3% of patients reported severe to extreme difficulty in doing these activities, and were not rated by 40.4% and 51.9% of patients, respectively. Those elements were followed by A1 "Descending stairs", A2 "Ascending stairs", and A4 "Standing" elements with a median of 3, in which 48.1%, 61.5%, and 59.6% of patients reported severe to extreme difficulty in these activities, in order. The median for the rest of the activities was between 1–2, in which 15.4–48.1% of patients reported severe to extreme difficulty.

In the KOOS-QoL subscale, the items' median ranged between 2–4. Patients reported the highest median when describing how often they are aware of their knee problem *(Q1. How often are you aware of your knee problem?)*, with 98.1% reporting "daily" or "constantly". Moreover, 69.2% of patients reported severe to extreme lack of confidence in their affected knee (Q3. *How much are you troubled with lack of confidence in your knee?*), and 75% reported severe to extreme general difficulty with their knee (Q4. *In general, how much difficulty do you have with your knee?*). However, 42.3% of participants have "severely" to "totally" modified their lifestyles to avoid potentially damaging activities (Q2. *Have you modified your lifestyle to avoid potentially damaging activities?).*

Hierarchical linear regressions revealed that knee pain on the less affected side significantly improved the model and explained an additional 19% of the variance in KOOS-ADL scores after accounting for age and BMI (Table 3). The addition of knee pain on the affected side (KOOS-Pain) also significantly improved the model and explained a further 50% of the

**Table 2. KOOS-subscales with the calculated median for each item, the percentage of patients rated the item with a specific Likert point, and percentage of patients who did not rate an item.**

| Subscale | Item | Median | (0) None % | (1) Mild % | (2) Moderate % | (3) Severe % | (4) Extreme % | Did not rate the item %, (N) |
|---|---|---|---|---|---|---|---|---|
| **Symptoms** | 1 | 2 | 36.5 | 5.8 | 26.9 | 13.5 | 17.3 | 0 |
| | 2 | 2 | 19.2 | 9.6 | 21.2 | 9.6 | 38.5 | 1.9% (1) |
| | 3 | 2 | 36.5 | 9.6 | 25 | 17.3 | 11.5 | 0 |
| | 4 | 0 | 75.0 | 1.9 | 1.9 | 9.6 | 9.6 | 1.9% (1) |
| | 5 | 4 | 15.4 | 5.8 | 7.7 | 5.8 | 55.8 | 9.6% (5) |
| | 6 | 2 | 26.9 | 3.8 | 32.7 | 26.9 | 9.6 | 0 |
| | 7 | 2 | 32.7 | 5.8 | 23.1 | 26.9 | 11.5 | 0 |
| **Pain** | 1 | 4 | 0 | 0 | 5.8 | 40.4 | 53.8 | 0 |
| | 2 | 3 | 5.8 | 5.8 | 15.4 | 17.3 | 19.3 | 36% (19) |
| | 3 | 1 | 32.7 | 21.2 | 13.5 | 15.4 | 9.6 | 7.7% (4) |
| | 4 | 3 | 13.5 | 13.5 | 9.6 | 11.5 | 26.9 | 25% (13) |
| | 5 | 2 | 19.2 | 21.2 | 28.8 | 23.1 | 7.7 | 0 |
| | 6 | 3 | 0 | 9.6 | 19.2 | 26.9 | 32.7 | 11.5% (6) |
| | 7 | 2 | 25.0 | 7.7 | 21.2 | 34.6 | 11.5 | 0 |
| | 8 | 1 | 34.6 | 21.2 | 30.8 | 11.5 | 1.9 | 0 |
| | 9 | 2 | 15.4 | 9.6 | 34.6 | 25.0 | 15.4 | 0 |
| **Function, daily living** | 1 | 3 | 1.9 | 7.7 | 36.5 | 23.1 | 25.0 | 5.8% (3) |
| | 2 | 3 | 0 | 7.7 | 26.9 | 32.7 | 28.8 | 3.8% (2) |
| | 3 | 2 | 15.4 | 13.5 | 34.6 | 26.9 | 7.7 | 1.9% (1) |
| | 4 | 3 | 7.7 | 1.9 | 30.8 | 48.1 | 11.5 | 0 |
| | 5 | 2 | 21.2 | 7.7 | 19.2 | 17.3 | 21.2 | 13.5% (7) |
| | 6 | 2 | 21.2 | 13.5 | 36.5 | 17.3 | 9.6 | 1.9% (1) |
| | 7 | 2 | 5.8 | 13.5 | 34.6 | 30.8 | 15.4 | 0 |
| | 8 | 4 | 0 | 1.9 | 7.7 | 17.3 | 32.7 | 40.4% (21) |
| | 9 | 1 | 36.5 | 15.4 | 25.0 | 15.4 | 1.9 | 5.8% (3) |
| | 10 | 2 | 13.5 | 17.3 | 28.8 | 23.1 | 17.3 | 0 |
| | 11 | 1 | 46.2 | 7.7 | 26.9 | 13.5 | 1.9 | 3.8% (2) |
| | 12 | 2 | 9.6 | 15.4 | 25.0 | 30.8 | 17.3 | 1.9% (1) |
| | 13 | 2 | 15.4 | 9.6 | 17.3 | 15.4 | 13.5 | 28.8% (15) |
| | 14 | 1 | 36.5 | 15.4 | 28.8 | 13.5 | 5.8 | 0 |
| | 15 | 2 | 28.8 | 19.2 | 28.8 | 13.5 | 9.6 | 0 |
| | 16 | 4 | 1.9 | 0 | 3.8 | 5.8 | 36.5 | 51.9% (27) |
| | 17 | 2 | 5.8 | 9.6 | 30.8 | 23.1 | 19.2 | 11.5% (6) |
| **Quality of Life** | 1 | 4 | 0 | 0 | 1.9 | 25.0 | 73.1 | 0 |
| | 2 | 2 | 9.6 | 23.1 | 25.0 | 32.7 | 9.6 | 0 |
| | 3 | 3 | 5.8 | 3.8 | 21.2 | 34.6 | 34.6 | 0 |
| | 4 | 3 | 3.8 | 1.9 | 19.2 | 53.8 | 21.2 | 0 |

variance in this measure (Table 3). Regarding the KOOS-QoL score, pain in the less affected knee did not contribute to this measure. While pain in the affected knee significantly predicted the KOOS-QoL score and explained 41% of the variance after accounting for variables in preceding steps (BMI and age, knee pain on the less affected side) (Table 3).

## Discussion

The goal of this study is to investigate the impact of end-stage knee OA on the perceived physical function and quality of life using the KOOS tool, and to examine the influence of knee pain

**Table 3. Hierarchical linear regression analysis with KOOS-ADLS and KOOS-QOL as dependent variables, and age, BMI, knee pain as independent variables.**

|  | R | $R^2$ | $R^2$ change | Significance of change | Significance of model |
|---|---|---|---|---|---|
| KOOS-ADLS Model |  |  |  |  |  |
| 1 | 0.252 | 0.063 | 0.063 | 0.214 | 0.214 |
| 2 | 0.503 | 0.253 | 0.189 | 0.001 | 0.004 |
| 3 | 0.869 | 0.755 | 0.502 | 0.000 | 0.000 |
| KOOS-QoL Model |  |  |  |  |  |
| 1 | 0.201 | 0.041 | 0.041 | 0.378 | 0.378 |
| 2 | 0.292 | 0.085 | 0.045 | 0.141 | 0.247 |
| 3 | 0.705 | 0.496 | 0.411 | 0.000 | 0.000 |

* **Model 1** = body mass index (BMI) and age; **Model 2** = BMI and age, pain in the "less affected" knee; **Model 3** = BMI and age, pain in the" less affected" knee, and pain in the "most affected" knee

on functional outcomes and QoL. Although several studies have examined the functional abilities of patients with knee OA in different countries, to our knowledge this is the first study conducted on the Jordanian population with end-stage knee OA.

Only four males took part in this study, which may reflect the fact that more women than men in Jordan have knee OA and are awaiting TKA. In a hospital-based cohort study in Jordan, women constitute 61.4% of patients diagnosed with severe radiographic knee OA [14]. Besides, in the USA, women are 40% more likely than men to develop knee osteoarthritis [18] and over 60% of TKA patients are females [8].

Findings of this study showed that the knee condition negatively impacts the functional abilities and quality of life as perceived by patients in the current study. In general, patients in this study scored low across KOOS subscales (28%-54%), indicating limited abilities in performing daily activities and poor quality of life. Although no population-based reference data for KOOS are available from the Jordanian population, patients in this study demonstrated poorer outcomes when compared to those reported in Southern Sweden [19], scoring 42%, 30%, 39%, and 60% lower in: pain, symptoms, ADL, and quality of life subscales, respectively (given that lower scores indicate worse outcomes). Regarding the patient population, various studies have evaluated the reported function by KOOS in those with knee OA [20, 21], and pre-operative to TKA [22–24]. The KOOS scores in the current study are slightly lower than those reported by Sivachidambaram et al. [20]; a study that examined the relationship between KOOS with the performance-based function in patients with primary Knee OA, with a mean difference ranging from 15 to 21 points across all KOOS subscales. Similarly, compared to the pre-operative KOOS scores reported in a study that evaluated the effect of preoperative physiotherapy on the functional outcomes after TKA [22], our scores were lower across all subscales, with a mean difference of **3–27** points. On the other hand, our patients' scores were slightly better compared to those reported by patients with symptomatic bilateral knee OA in Sabirli and colleagues study [21], with a mean difference of 1–10 points across KOOS subscales. Moreover, when compared to the scores reported by Rastogi [24], a study conducted on patients before undergoing primary TKA, our scores were very similar to theirs.

In the current study, each subscale's element and its difficulty level have been thoroughly examined. Patients in our sample reported that the most impairment is in their ability to bend their knee fully. Moreover, most patients (94.2%) experienced knee pain that persists daily or always. When looking at the elements of the daily activities' subscale, there were dissimilar responses across the elements indicating that some activities are more difficult to perform than others. The most challenging activities were when going shopping and doing heavy domestic

duties, followed by ascending and descending stairs and standing. It is noticeable that 50% and 42.3% of responders reported: "difficulty to extreme difficulty" when going shopping and performing heavy domestic duties, in order. At the same time, a large proportion of patients (40.4% and 51.9%, respectively) did not rate the difficulty they encountered when performing these two activities, which could be due to their inability or unwillingness to perform these tasks. Patients in current study are older adults; in our culture, it is uncommon for the elderly to engage in such activities. Moreover, ascending stairs, standing, and descending stairs activities seem challenging in which 61.5%, 59.6% and 48.1% of patients recognized their performance as difficult to extremely difficult. On the other hand, some tasks are the easiest for our patients including "Putting on socks", "Taking off socks", and "sitting". Overall, our patients expressed difficulties in tasks that are required daily.

Regarding the impact on health related QoL, osteoarthritis has a strong clinically important negative effect [25], and patients with knee OA have significantly lower QoL compared with healthy controls [26, 27]. Our findings were similar; patients scored very low on the QoL subscale. Nearly all patients in this study reported they are aware of their knee condition on a daily or ongoing basis, and most of them are severely to extremely troubled because they lack confidence in their knee and have severe to extreme general difficulty with their knee. This indicates how the knee OA condition mostly influences not only the physical but also the emotional functioning and the level of confidence as measured by the KOOS-QoL subscale. Interestingly, after excluding the sport and recreation subscale, the findings of this study are consistent with a general trend observed in earlier studies with the QoL subscale having the lowest score [28–30], and the symptom subscale having the maximum score [28, 29] among the KOOS subscales.

According to the findings of this study, knee pain on the "less affected" and on the "most affected" sides were determinants of self-perceived function on the KOOS-ADLs. While only pain in the "most affected" side was a determinant for the knee related QoL. These results are consistent with those reported in previous studies; joint pain was a predictor to self-reported function in patients with knee OA [31–33], and in those with hip OA [34]. Mizner and colleagues also reported that pain was strongly related to the patient-reported KOS-ADLs score [35]. Besides, a study by Stonga et al revealed that patients with end-stage knee OA are at greater risk of falling compared to healthy older adults, with pain and limited functional ability negatively influencing the patient's quality of life and increased the fall risk [36].

It is expected that functional abilities and QoL in our sample will continue to worsen during the waiting period. It has been shown that patients with knee OA demonstrated a reduced quality of life compared to the age-matched norms, and as the disease progresses, the QoL declines [27, 37]. It was reported by Ackerman and colleagues that health-related QoL continues to deteriorate in 75% of patients awaiting for TKA during the mean 10-month waiting period before surgery [37]. Moreover, patients waiting for total hip and knee arthroplasty perceived the surgical wait as a contributor to the amount of health deterioration [38]. It is clearly evident that primary objective in OA rehabilitation is to reduce joint pain, increase strength, and improve the functional abilities and quality of life [39]. Given that pain is a determinant of functional ability and QoL, our findings may emphasize the importance and benefit of providing a therapeutic intervention to help relieve or control osteoarthritis-related knee pain which may mitigate the potential deterioration in functional and QoL levels during waiting time. Patients in the current study, however, reported no exercise or physical therapy intervention while awaiting TKA.

Lifestyle modification, such as regular low-impact exercise, mass reduction, and avoidance of activities that significantly aggravate the patients' symptoms, is considered a core part of the conservative treatment for knee OA [40, 41]. These modifications can help in reducing knee

pain and slowing the progression of the condition [41]. We noticed that 33% of patients in current study either did not or only slightly modified their lifestyle while 42% of patients have severely to totally modified their lifestyle (Table 2, QoL element 2). This variability in responses may be attributed to the patient's level of pain; the severity of the knee pain may necessitate lifestyle modifications. However, it may also indicate a lack of awareness about the importance of adopting such modifications to minimize further potential damage to their knees. It was reported by Al-Khlaifat et al. that Jordanian patients with knee OA primarily rely on medication as first option to manage their pain, which may be attributed to several factors including a lack of knowledge about pain management, a lack of education on the importance of therapeutic exercises, ineffective service delivery process, and inappropriate exercise prescription [42]. Hence, increasing patient's awareness about pain management and lifestyle modifications to reduce pain is warranted. It is of utmost importance to encourage the patient to participate in rehabilitation program to reduce pain and improve function while waiting for TKA which may enhance/ or slow down the deterioration their functional and QoL outcomes, with physicians, surgeons, and therapists having a crucial role in doing so. Moreover, it is a message to the orthopedic surgeons to consider the patient's perceived functional outcomes in clinical decision-making when determining the timing and priority of TKA.

## Limitations and future directions

This study in not without any limitations. Our sample may not be representative of all patients with end-stage knee OA across Jordan. Even though knee OA is more prevalent among female patients, the overrepresentation of females in our study participants and the exclusion of comorbid conditions that are common in this age group may limit the generalizability of our findings to all individuals with knee OA. The effect of gender on KOOS outcome couldn't be evaluated in this study due to the limited number of male participants, and this could be investigated in a future study with a more diverse and representative sample. Additionally, in this analysis, we looked at the relation between pain with the perceived function and quality of life, however; many other variables could also be considered as contributing factors but could not be included in this study, such as muscle strength, socioeconomic status, psychological factors, disease duration of knee OA, and level of physical activity. However, it was not possible in this study to cover all the complex variables influencing knee related perceived function and quality of life. Future studies that investigate the influence of such variables on knee-related perceived function and quality of life, could provide a more comprehensive understanding of the subject, which could, in turn, inform the development of more effective interventions for patients with end-stage knee OA.

## Conclusion

Although our findings are parallel to previous studies from different countries, an accurate comparison cannot be made due to different inclusion criteria and severity of the knee condition on those studies. Overall, our study indicates that Jordanian patients with end-stage knee OA showed a substantial impairment in perceived function and knee-related quality of life. The perception of functional abilities and quality of life is negatively influenced by knee pain level in our study group. This study emphasizes the importance of addressing knee pain through a targeted regimen prior to TKA, as well as raising patient awareness about pain management and the importance of exercise, which may positively affect the perceived functional ability and quality of life while waiting for TKA.

## Supporting information

**S1 File Dataset. .**
(XLSX)

## Acknowledgments

We acknowledge the research assistant; Lubna Alnajjar for her contribution in recruiting subjects and collecting data. We would like to thank the University of Jordan Hospital, Department of Orthopaedics, and particularly the referring surgeons for their continued support for this work. We also thank all the participants in this study.

## Author Contributions

**Conceptualization:** Sumayeh Abujaber, Ibrahim Altubasi.

**Data curation:** Sumayeh Abujaber, Ibrahim Altubasi.

**Formal analysis:** Sumayeh Abujaber, Ibrahim Altubasi.

**Funding acquisition:** Sumayeh Abujaber.

**Investigation:** Sumayeh Abujaber.

**Methodology:** Sumayeh Abujaber, Ibrahim Altubasi.

**Project administration:** Sumayeh Abujaber.

**Resources:** Mohammad Hamdan, Raed Al-Zaben.

**Writing – original draft:** Sumayeh Abujaber, Ibrahim Altubasi.

**Writing – review & editing:** Sumayeh Abujaber, Ibrahim Altubasi, Mohammad Hamdan, Raed Al-Zaben.

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
