## [Decision Letter · Decision Letter 0]

15 Mar 2023

PONE-D-22-33473Impact of end-stage knee osteoarthritis on perceived physical function and quality of life in the Middle East: A descriptive study in JordanPLOS ONE

Dear Dr. Abujaber,

Thank you for submitting your manuscript to PLOS ONE. After careful consideration, we feel that it has merit but does not fully meet PLOS ONE’s publication criteria as it currently stands. Therefore, we invite you to submit a revised version of the manuscript that addresses the points raised during the review process.

ACADEMIC EDITOR:Revise according to the reviewers' comments and clearly highlight potential limitations for this study and future directions. 

We look forward to receiving your revised manuscript.

Kind regards,

Aqeel M Alenazi

Academic Editor

PLOS ONE

Additional Editor Comments:

Revise according to the reviewers' comments and clearly highlight potential limitations for this study and future directions.

Reviewers' comments:

Reviewer's Responses to Questions

**Comments to the Author**

1. Is the manuscript technically sound, and do the data support the conclusions?

Reviewer #1: Yes

Reviewer #2: Partly

Reviewer #3: Yes

2. Has the statistical analysis been performed appropriately and rigorously? 

Reviewer #1: Yes

Reviewer #2: N/A

Reviewer #3: Yes

3. Have the authors made all data underlying the findings in their manuscript fully available?

Reviewer #1: No

Reviewer #2: No

Reviewer #3: Yes

4. Is the manuscript presented in an intelligible fashion and written in standard English?

Reviewer #1: Yes

Reviewer #2: No

Reviewer #3: Yes

5. Review Comments to the Author

Reviewer #1: Nice work! I would suggest to mention about how the Knee Injury and Osteoarthritis Outcome Score (KOOS) was translated to Arabic. Having only 4 male out of 52 participants is not optimal. Also, I am not sure if you calculated the power analysis. There were no justification about why you used that normalization method for KOOS.

Reviewer #2: İn line 97-101

Jordanians have cultural habits that involve cross-sitting and kneeling for lengthy periods during the day, most of them are Arab Muslims who pray five times a day, and this religious activity includes kneeling, knee bending, and

frequent squatting which might exacerbate the loads on the knees. They also tend to postpone surgical intervention decisions. ( This sentence is just your opinion. Not well-intentioned also. The literature says the opposite. You haven't even looked at the literature. You can look this study, Yılmaz, S., Kart-Köseoglu, H., Guler, O., & Yucel, E. (2008). Effect of prayer on osteoarthritis and osteoporosis. Rheumatology International, 28, 429-436. )

Reviewer #3: 1. Author need to revised the title as in the title it is written middle east, whereas study has only included the participants from only one hospital with sample of 52 only. So this can't be generalize to whole middle east.

2. It will be good to add end-stage knee OA in the short title.

3. It will be good to remove KOOS in the keyword or write the full term.

4. see the referencing style.

5. Result: Line 198-200: Doesn't make clear sense. Write it clearly.

6. Limitation: The major limitation of the study is limited sample size and majority are female. So, the findings can't be generalize to larger population. Although OA knee is common among female population.

6. PLOS authors have the option to publish the peer review history of their article (what does this mean?). If published, this will include your full peer review and any attached files.

Reviewer #1: No

Reviewer #2: **Yes: **Esedullah AKARAS

Reviewer #3: **Yes: **Bishnu Dutta Acharya

---

## [Author Response · Author response to Decision Letter 0]

22 Apr 2023

Dear Dr. Alenazi, 

I would like to thank you for the opportunity to revise and resubmit our manuscript (PONE-D-22-33473) entitled “Impact of end-stage knee osteoarthritis on perceived physical function and quality of life in the Middle East: A descriptive study in Jordan.” We would also like to express our thanks to the reviewers for their comments, and we greatly appreciated the constructive criticisms in evaluating our manuscript.

We fully believe that the reviewers' comments are helpful in improving the manuscript, and we have carefully considered and responded to each suggestion. The uploaded "response to reviewers" document outlines our efforts to address the reviewers' comments and concerns on a point-by-point basis. We hope that our responses and the revised manuscript are satisfactory and sufficiently address your concerns about the study.

Regarding the data availability, I have uploaded the minimal anonymized dataset as a supporting information file. 

Thank you again for your consideration of our revised manuscript.

Sincerely yours,

Corresponding author:

Sumayeh Abujaber. PT, PhD

S.abujaber@ju.edu.jo

---

## [Decision Letter · Decision Letter 1]

29 May 2023

Impact of end-stage knee osteoarthritis on perceived physical function and quality of life: A descriptive study from Jordan

PONE-D-22-33473R1

Dear Dr. Abujaber,

We’re pleased to inform you that your manuscript has been judged scientifically suitable for publication and will be formally accepted for publication once it meets all outstanding technical requirements.

Kind regards,

Aqeel M Alenazi

Academic Editor

PLOS ONE

Additional Editor Comments (optional):  The authors have addressed all the comments suggested by the reviewers. The concern regarding sample size has been sufficiently addressed in the limitations section of the manuscript.

Reviewers' comments:

Reviewer's Responses to Questions

**Comments to the Author**

1. If the authors have adequately addressed your comments raised in a previous round of review and you feel that this manuscript is now acceptable for publication, you may indicate that here to bypass the “Comments to the Author” section, enter your conflict of interest statement in the “Confidential to Editor” section, and submit your "Accept" recommendation.

Reviewer #2: All comments have been addressed

Reviewer #3: All comments have been addressed

2. Is the manuscript technically sound, and do the data support the conclusions?

Reviewer #2: Yes

Reviewer #3: Partly

3. Has the statistical analysis been performed appropriately and rigorously? 

Reviewer #2: N/A

Reviewer #3: Yes

4. Have the authors made all data underlying the findings in their manuscript fully available?

Reviewer #2: Yes

Reviewer #3: Yes

5. Is the manuscript presented in an intelligible fashion and written in standard English?

Reviewer #2: Yes

Reviewer #3: Yes

6. Review Comments to the Author

Reviewer #2: All comments have been addressed.

Reviewer #3: The author has written the paper nicely however, to consider for the publication there should be at least 100 sample size so the findings of study can be generalized.

7. PLOS authors have the option to publish the peer review history of their article (what does this mean?). If published, this will include your full peer review and any attached files.

Reviewer #2: **Yes: **Esedullah AKARAS

Reviewer #3: **Yes: **Bishnu Dutta Acharya

---

## [Editor Report · Acceptance letter]

2 Jun 2023

PONE-D-22-33473R1 

Impact of end-stage knee osteoarthritis on perceived physical function and quality of life: A descriptive study from Jordan 

Dear Dr. Abujaber:

I'm pleased to inform you that your manuscript has been deemed suitable for publication in PLOS ONE. Congratulations! Your manuscript is now with our production department. 

Kind regards, 

on behalf of

Dr. Aqeel M Alenazi 

Academic Editor

PLOS ONE